**RESEARCH**

# Conserved missense variant pathogenicity and correlated phenotypes across paralogous genes

Tobias Brünger[1,4†], Alina Ivaniuk[2†], Eduardo Pérez-Palma[3], Ludovica Montanucci[4], Stacey Cohen[5,6,7], Lacey Smith[8], Shridhar Parthasarathy[5,6,7], Ingo Helbig[5,6,7,9], Michael Nothnagel[1], Patrick May[10] and Dennis Lal[1,4,11,12,13*]

[†]Tobias Brünger and Alina Ivaniuk contributed equally to this work.

*Correspondence:
dennis.lal@uth.tmc.edu

[1] Cologne Center for Genomics (CCG), University of Cologne, Cologne 50931, Germany
Full list of author information is available at the end of the article

## Abstract

**Background:** The majority of missense variants in clinical genetic tests are classified as variants of uncertain significance. Prior research shows that the deleterious effects and the subsequent molecular consequences of variants are often conserved among paralogous protein sequences within a gene family. Here, we systematically quantify on an exome-wide scale whether the existence of pathogenic variants in paralogous genes at a conserved position can serve as evidence for the pathogenicity of a new variant. For the gene family of voltage-gated sodium channels, where variants and expert-curated clinical phenotypes are available, we also assess whether phenotype patterns of multiple disorders for each gene are conserved across variant positions within the gene family.

**Results:** Mapping 590,000 pathogenic and 1.9 million population variants onto 9928 genes grouped into 2054 paralogous families increases the number of residues with classifiable evidence 5.1-fold compared with gene-specific data alone. The presence of a pathogenic variant in a paralogous gene is associated with a positive likelihood ratio of 13.0 for variant pathogenicity. Across ten genes encoding voltage-gated sodium channels and 22 expert-curated disorders, we identify cross-paralog correlated phenotypes based on 3D structure spatial position. For example, multiple established loss-of-function related disorders across SCN1A, SCN2A, SCN5A, and SCN8A show overlapping spatial variant clusters. Finally, we show that phenotype integration in paralog variant selection improves variant classification.

**Conclusion:** Conserved pathogenic missense variants in paralogous genes provide robust, quantifiable support for clinical variant interpretation, and phenotype-informed mapping further improves predictions.

**Keywords:** Genetics, ACMG, variant classification, missense, sodium channel

## Background

Large gene panels, exome, and genome sequencing have led to the identification of novel variants at an exponential rate [1]. Up to 80% of pathogenic variants are located within protein-coding regions of the gene [2], with missense variants being particularly challenging to interpret due to the variety of different molecular mechanisms through which they can cause disease. Furthermore, several disease-associated genes are pleiotropic, further complicating variant interpretation [3–5]. Despite these challenges, variant classification is necessary for diagnosing rare and genetically heterogeneous disorders, and for the development of personalized medicine.

About 80% of genes associated with monogenic disorders are paralogs [6]. These paralogous genes can be grouped into 2871 gene families as defined by the Human Gene Nomenclature Consortium (HGNC) [7] with > 80% sequence similarity [8]. Genes within a gene family arise from gene duplication events of common ancestral genes and can share > 90% amino acid sequence similarity at functionally essential protein domains [9]. We and others have shown that quantifying conservation across these paralogous genes and homologous domains is an effective strategy to distinguish between pathogenic and benign variants [8, 10–12]. Molecular studies further indicate that the biophysical function of domains is conserved within a gene family. As a result, a single amino acid substitution in the same position of a homologous domain often leads to similar molecular effects across members of the same gene family [11, 13, 14]. This suggests that a comprehensive understanding of variants in one gene can provide, through a form of knowledge "transfer," insights into the pathogenicity and also into the biological disease mechanisms of unstudied variants in its paralogs when these variants are located at identical positions.

Within the same gene family, proteins show similar patterns of population variant-constrained and pathogenic variant clustering. In addition to identifying conservation patterns within gene families, previous research has highlighted the differential distribution of missense variants between the general population and pathogenic missense variants which was consistent across a subset of paralogous genes [11, 14]. Furthermore, our previous findings indicate that this regional clustering is prevalent across paralogous genes and enables a systematic identification of regions enriched with pathogenic variants, termed pathogenic variant enriched regions (PERs) [15]. Our study showed that novel missense variants located within PERs have a higher likelihood of being pathogenic compared to those in non-PER regions of the same gene [15]. However, this method currently has limited sensitivity, since many newly discovered variants are located outside of PERs. Moreover, as PERs typically define a larger protein region, interpretations regarding disease mechanisms are constrained to a regional context, preventing insights at the individual amino acid level.

To standardize variant interpretation, the American College of Medical Genetics and Genomics (ACMG) and the Association for Molecular Pathology (AMP) published recommendations for evaluating the pathogenicity of variants [16]. However, > 45% of single nucleotide variants reported in the ClinVar database [17] (accessed March 2023) are classified as variants of uncertain significance (VUS), due to the absence of sufficient evidence for or against variant pathogenicity. The guidelines include criteria that utilize information from previous variant classifications, e.g., the presence of an established

pathogenic variant with the same amino acid exchange (PS1) or a different amino acid exchange (PM5) at the same position in the same gene that can provide strong to moderate evidence for pathogenicity [16]. However, since the vast majority of rare monogenic disorders are genetically heterogeneous and about half of the identified pathogenic variants have not yet been observed in other individuals [18, 19], the application of these evidence criteria is limited.

In the present study, we extend prior work on gene family conservation to provide access to a paralog-based annotation that could improve the assessment of variant pathogenicity. We postulate that variants previously classified in conserved residues of paralogous genes can provide evidence for the pathogenicity of novel variants located at corresponding amino acid positions in these genes. The use of pre-classified variants in paralogs as evidence of pathogenicity has been previously suggested for a select group of genes, e.g., by the RASopathy ClinGen Expert Panel [20]. However, the broad applicability of this approach across the entire protein-coding exome—particularly, the potential of single missense variants from paralogs as a feature to inform variant pathogenicity—remains unquantified and untested.

In this proof of concept study, our findings reveal that for 414 gene families (comprising 2193 genes) with high sequence similarity, the presence of a pathogenic variant in one gene family member at an equivalent protein position is associated with a significant increase in the likelihood of pathogenicity for a novel variant at a conserved paralogous site in the target gene. Additionally, we illustrate in a case study that integrating expert-curated clinical data across sodium channels can refine variant selection, which not only enhances variant pathogenicity classification but also identifies disorders across paralogs that likely share similar disease mechanisms.

## Results

### Incorporating pathogenic paralogous variants triples classifiable amino acid residues

The guidelines of the ACMG suggest that for determining the pathogenicity of novel variants, two scenarios can be considered: (1) the presence of a variant in the same gene with an identical amino acid change, irrespective of the nucleotide alteration and (2) a novel amino acid substitution at a position where another substitution was previously been considered pathogenic, named PS1 and PM5 criteria respecitvely [16]. In this study, our objective was to explore whether this principle could be extrapolated to encompass pathogenic variants in paralogous genes. We specifically assessed if the existence of pathogenic variants in paralogous genes at a conserved, corresponding position could serve as evidence for the pathogenicity of a new variant. For our study, we termed a "paralogous variant" as a variant that meets two conditions: (1) it is positioned in a paralogous gene at the analogous residue index position, as delineated by multiple sequence alignment (refer to the " Methods" section for details), and (2) it shares the same reference amino acid as the target gene.

First, we assessed the number of amino acid residues not overlapping with pathogenic variants within the same gene at equivalent paralogous amino acid positions, but yet overlapping with pathogenic variants in paralogous genes. We aggregated a total of 38,321 pathogenic variants from ClinVar [17] and HGMD [21] and mapped them to 2840 different gene family alignments, consisting of 9928 genes (Fig. 1). Our paralog variant

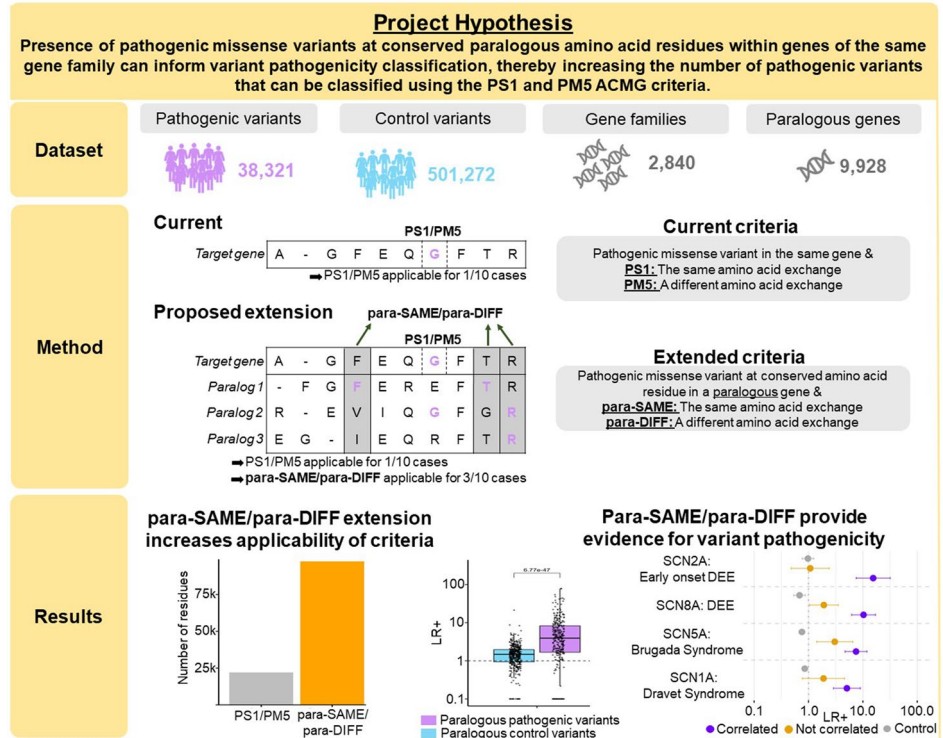

**Fig. 1** Graphical summary of the study

analysis integrates pathogenic variants from multiple genes in the same gene family (see the " Methods" section for details). We, therefore, restricted the dataset to gene families harboring pathogenic variants in at least two genes and identified 414 gene families. Within these genes, 30,117 pathogenic missense variants and 140,208 pathogenic paralogous variants were found that covered 22,071 and 67,273 amino acid residues respectively (Table S1). Of these 67,273 residues that are covered by a paralogous pathogenic variant 91.7% ($N = 61,670$ residues) were not covered by a pathogenic variant in the same gene. Therefore, the integration of paralogous pathogenic variants would increase the number of amino acids in these gene families were the criteria can be applied by about 2.8-fold ($N = 83,741$ residues, Fig. 2A). The increase in the number of classifiable amino acids in each gene family is highly correlated with the number of disease-associated genes in a gene family ($R = 0.96$, $P = 2.4e - 9$, Additional file 1: Fig. S1).

### Presence of single pathogenic paralogous variants can be used to assess variant pathogenicity

Next, we quantified the value of incorporating pathogenic variants at paralogous positions to assess the variant pathogenicity of novel variants. Therefore, in addition to the aforementioned pathogenic variants, we included 550,251 variants from the Regeneron One Million Exome database [22] that were not present in gnomAD [23], which served as controls in our study. When a pathogenic paralogous variant with the same amino acid exchange was present at a corresponding alignment index position (termed the para-SAME criterion, for details on the approach, see the " Methods" section), we

observed across 414 gene families an average LR + of 13.0 (12.5–13.7, 95% confidence interval [CI]; Fig. 2B). Notably, even for paralogous variants with a different substitution at the same alignment index position (termed the para-DIFF criterion), we observed an LR + (LR + = 6.0 [5.7–6.2, 95% CI). We observe that LR + values remain highly correlated across different variant set compositions, demonstrating the robustness of our approach, while considering variability in classification quality of pathogenic and control variants (Additional file 1: Fig. S2). However, we observed substantial variation in the observed LR + across different genes (Fig. 2B), indicating that gene-family-specific calibration is advisable. We further investigated whether the presence of multiple pathogenic paralogous variants at the same conserved residue increases the evidence for pathogenicity. As shown in Additional file 1: Fig. S3, a single pathogenic variant in a paralogous gene is associated with a 2.84-fold enrichment of pathogenic variants compared to benign variants at that residue. The fold enrichment of pathogenic variant progressively increases with a higher number of pathogenic paralogous variants.

### The presence of pathogenic paralogous variants provides evidence for pathogenicity beyond evolutionary conservation

Variant mapping across paralogous residues requires residue conservation. Next, we investigated the added value of mapping beyond conservation. Previously, we developed a "parazscore [8]" to measure the conservation across paralog genes, showing that amino acids conserved within a gene family are significantly enriched for pathogenic variants. Notably, a fundamental prerequisite for the incorporation of pathogenic paralogous variants into the variant is the conservation of amino acid residues between the target gene and its paralogous gene. Hence, whenever pathogenic paralogous variants criteria are incorporated, a certain degree of conservation within the genes of the same gene family becomes inevitable. This conservation likely explains a portion of the elevated LR + we observed. Notably, while many methods [24–26] employ evolutionary conservation as a predictor of variant pathogenicity, it is crucial to discern the added value our approach provides beyond solely relying on conservation-based evidence. To achieve this, we reconsidered our previous analysis, segmenting amino acids based on their paralog conservation and grouping amino acid residues with similar conservation across paralogs together (see the " Methods" section for details). Interestingly, within these subgroups, the highest LR + were observed for residues exhibiting the least paralog conservation for both the para-SAME criterium (Parazscore < 0; $LR+_{para-SAME} = 25.2$, 95% CI = 21.7–29.3, Fig. 3A) as well as the para-DIFF criterium (Parazscore < 0; $LR+_{para-DIFF} = 9.7$, 95% CI = 8.7–10.7, Fig. 3B). Yet, even within the subgroup demonstrating the least increase in LR +, where maximum conservation across all paralogous genes of the same gene family was noted, we still detected an increased LR + of 5.8 and 2.9, for para-SAME and para-DIFF criteria respectively. This observation suggests that the existence of pathogenic paralogous variants provides additional information beyond the level of conservation between paralogous genes.

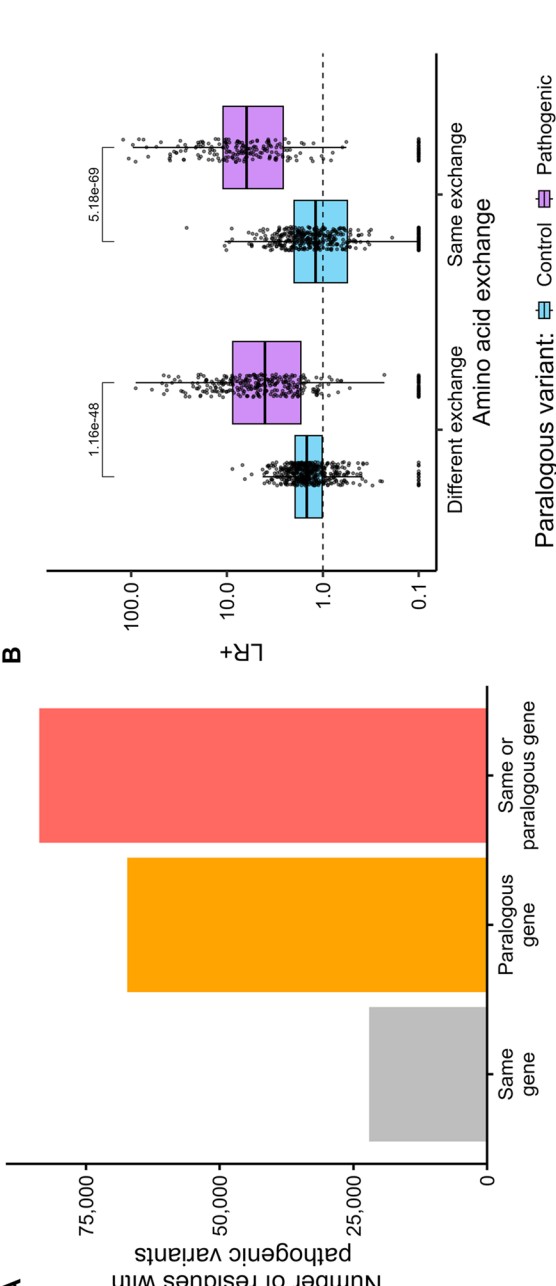

**Fig. 2** Individual pathogenic paralogous variants can serve as a proxy for variant pathogenicity. **A** Number of amino acid residues in 414 gene families that have a pathogenic variant (ClinVar, one star+) at the same protein position in the same gene or a corresponding protein residue in a paralogous gene. **B** Amino acids with a paralogous pathogenic variant at a paralogous aliment position have an increased positive likelihood ratio (LR+ > 1). In contrast, amino acids with a paralogous control variant (Regeneron, not present in gnomAD) at a paralogous alignment position are not enriched for pathogenic variants (Regeneron, not present in gnomAD). Each data point represents the gene-wise LR+. The gene-wise LR+ was calculated for genes where 10 or more pathogenic variants (ClinVar, one star+)) and control variants (Regeneron, not present in gnomAD) could be mapped

### Integrating single pathogenic paralogous variants improves a previous family-based variant interpretation approach

We compared our approach, using paralogous pathogenic variants located at corresponding amino acids to a previously published method [15]. In contrast to our new approach, the published approach identifies "pathogenic variant enriched regions" ("PERs," on average 33 consecutive amino acids [15]) across a gene family that is consistently enriched for pathogenic variants while depleted for control variants. Due to the sliding window approach the identified regions that are enriched for pathogenic variants, PERs can span amino acid residues without an established pathogenic variant across paralogs, and the regional association is derived from adjacent variants. However, identifying PERs within a gene family alignment requires a large number of pathogenic variants, limiting its applicability. First, we compared the number of exome-wide classifiable variants using single paralogous pathogenic variants with the PER approach. We used an independent set of pathogenic and control variants that were not utilized in the PER generation or the application of the para-SAME/DIFF criteria (see the " Methods" section for details). We found that the approach based on single paralogous pathogenic variants captured 2.9 times more residues compared to PERs (Fig. 3C). In the second comparison, we compared the LR + for each approach and observed a LR + 5.7 (5.4–6.0 95% CI) for the PER approach compared to a LR + of 8.6 (8.0–9.3 95% CI) for the para-SAME approach (Fig. 3D).

### Leveraging phenotype correlations across paralogs can enhance pathogenicity assessment

A single gene can be associated with different disorders. The number of disorders associated with variants in the same gene frequently correlates with the number of different molecular functional defects. Given that structure determines function, the molecular consequences of variants often relate to their specific position within the protein structure [27]. Thus, pinpointing phenotype correlations based on analogous variant distributions might reveal paralogous variants with consistent molecular effects. In the context of voltage-gated sodium channels (VGSCs), past research has underscored not only the conservation of pathogenicity but also the consistent functional effects among paralogous variants [13]. Building on this, we hypothesized that uncovering phenotype correlations across VGSCs could fine-tune the application of pathogenic paralogous variants for variant pathogenicity assessment. We hypothesize that within gene family phenotype correlations could identify correlated phenotypes based on substitution position, subsequently enhancing the likelihood of conserved pathogenicity for variants at equivalent positions. To test this hypothesis, we curated a comprehensive dataset featuring 945 affected individuals, associated with 22 diverse phenotypes and possessing 886 unique missense variants in VGSC-encoding genes (detailed in Table S2). Performing alignment position-based mapping onto the same structure combined with spatial-based phenotype proximity correlation analysis (see the " Methods" section for details), we identified within gene family position correlated phenotypes (Fig. 4A). For example, *SCN1A*-associated Dravet syndrome variants exhibited 3D positional correlations with *SCN2A* variants associated with autism ($R = 0.31$, $P = 4.7e - 35$), and Brugada syndrome variants in *SCN5A* ($R = 0.29$, $P = 6.1e - 40$).

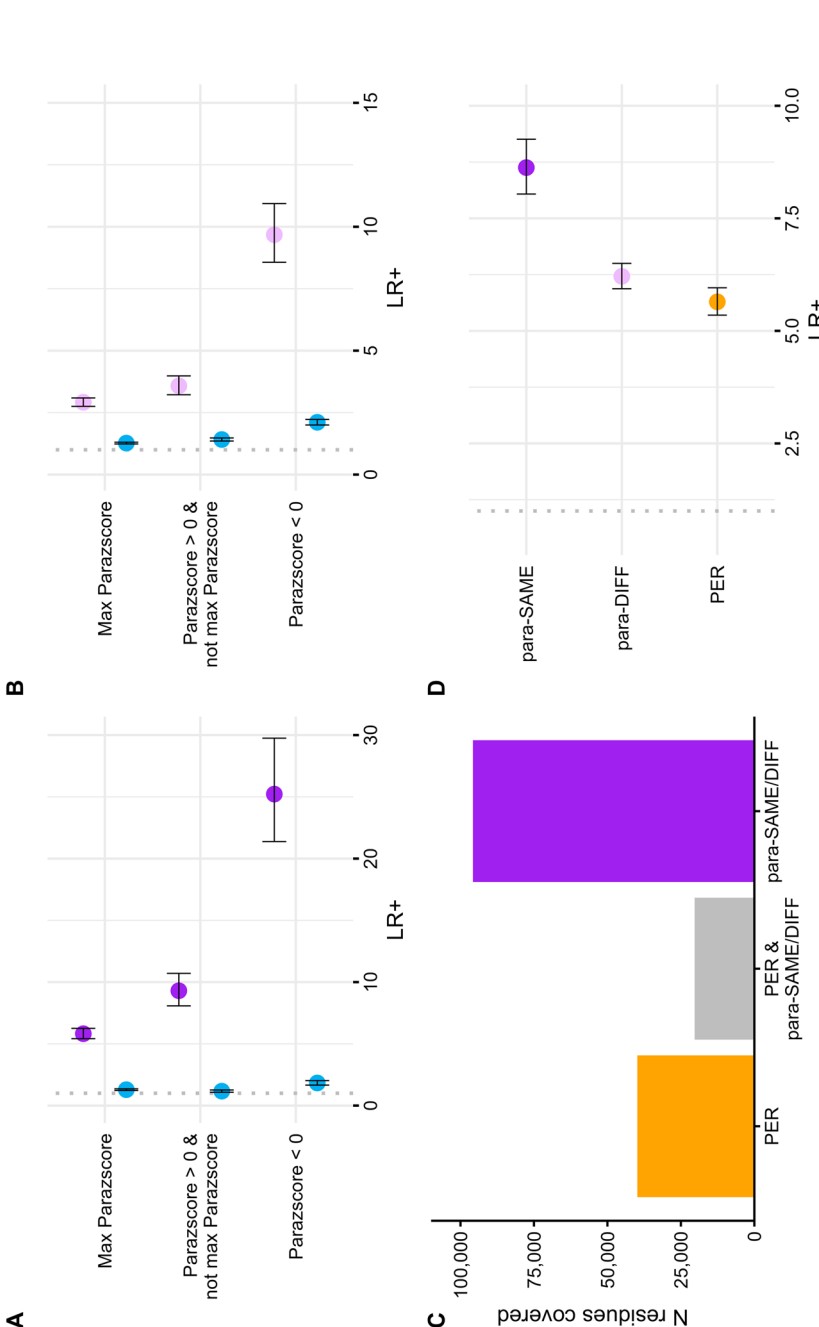

**Fig. 3** Comparison to established gene family-based methods. **A** The forest plot illustrates the enrichment of pathogenic versus control variants applying the para-SAME criterium for residues with similar paralog conservation levels, as defined in Lal et al. (2020) [8]. **B** Similar to **A**, but for the para-DIFF criterium. **C** The bar plot shows the number (*N*) of amino acid residues across all genes where a previously established approach (Pathogenic Enriched Region, PER; Perez-Palma et al., 2019 [15]) and/or our para-SAME/para-DIFF ACMG criteria extension can be applied. **D** The forest plot compares the likelihood ratios (LR+) for amino acid residues within a PER and amino acid residues where para-SAME/para-DIFF criteria can be applied (see the " Methods" section for details)

For genes associated with several related disorders, such as the VGSC, variant classification is challenging since phenotype specificity is not high. Therefore, not all pathogenic classified variants might be correctly classified. Next, we tested whether variants from spatially correlated phenotypes across different paralogous genes could increase variant pathogenicity classification accuracy. We selected the most frequently reported phenotypes for VGSC genes with at least 40 patients. The four genes *SCN1A*, *SCN2A*, *SCN5A*, and *SCN8A* fulfilled this criterion. We dissected the associated variants into four subsets and calculated the evidence for variant pathogenicity (see the " Methods" section for details). We observed an increased positive likelihood ratio by a factor of 3.6–14.2 for paralogous variants associated with 3D-position correlated phenotypes, in contrast to those paralogous variants without a significant 3D-position correlation (Fig. 4B). For example, for SCN8A DEE cases pathogenic paralogous variants whose phenotype correlate with the DEE in SCN8A (LR+ = 10.5, CI 6.4–17.1) showed an 3.7-fold higher strength to assess variant pathogenicity compared to pathogenic paralogous variants found in cases with non-correlating phenotypes (LR+ = 2.81, CI 1.4–5.8).

## Discussion

Many paralogs are highly conserved in sequence and have similar biophysical molecular functions. Current variant interpretation guidelines only consider previously classified pathogenic missense variants in the gene of interest as evidence for pathogenicity. Here, we developed and validated a bioinformatic framework to integrate pathogenic missense variants in paralogous genes at corresponding alignment index positions as evidence for the pathogenicity of novel variants. We demonstrated that integrating paralogous pathogenic variants located at a corresponding protein position can provide evidence for pathogenicity even if the amino acid exchange is not conserved. Compared to approaches, such as the PS1 and PM5 criteria of the ACMG guidelines [16] which consider pathogenic variants in the same gene at the same position as evidence, our approach can be applied to 5.1-fold more protein residues where novel variants of unknown pathogenicity could be observed.

Pathogenic missense variants in paralogous genes can serve as a proxy for pathogenicity. Within a protein sequence, pathogenic variants are unevenly distributed and tend to accumulate in certain regions that are critical for protein function [28]. These pathogenic variant-enriched regions have proven valuable for variant classification through established guidelines for variant interpretation [16] and the use of in silico prediction algorithms [29]. Moreover, the observation that critical protein regions tend to be evolutionarily conserved between paralogous genes can be harnessed to enhance statistical robustness by incorporating pathogenic variants across these paralogous genes [15]. Still, about 70%, of pathogenic variants are located outside the regions identified as essential. As a result, individual pathogenic variants in paralogous genes outside these regions were not considered for variant interpretation. In a study examining long QT syndrome, it was observed that individual pathogenic variants in paralogous genes are often located at paralogous positions as determined from multiple sequence alignments [11], suggesting that the presence of a pathogenic variant at a particular position may serve as a proxy of pathogenicity at that alignment position in other paralogs. While pathogenic paralogous variants provide strong evidence for pathogenicity, we showed

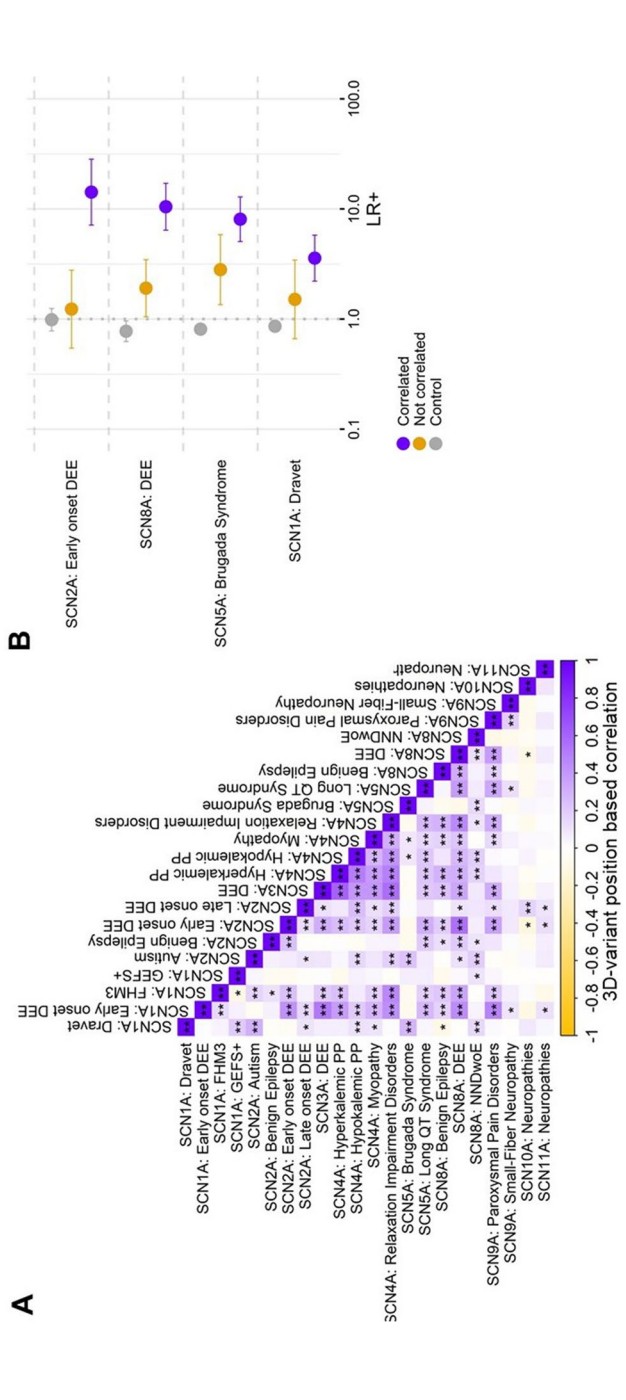

**Fig. 4** Leveraging phenotype correlations to enhance the application of paralogous pathogenic variants. **A** Displayed is a correlation matrix that delineates the relationships between the 3D variant distributions across various phenotypes. Phenotypes that share significantly (after Bonferroni adjustment) similar 3D-variant distributions are color-coded in purple, whereas those with significantly distinct distributions are in orange. Statistically significant correlations are marked with stars (* for *P*adj < 0.05 and ** for *P* < 0.001). **B** Presented is a forest plot capturing the positive likelihood ratio for four pivotal phenotypes that is derived from a comparison of affected individuals and control variants sourced from gnomAD. These ratios were computed by either (1) employing paralogous variants from affected individuals that exhibited a significantly positive correlation on 3D position (depicted in purple), (2) utilizing paralogous variants from affected individuals displaying a 3D position-based negative correlation (showcased in orange), and (3) considering paralogous control variants (represented in gray). Abbreviations: DEE, Developmental Epileptic Encephalopathy; Dravet, Dravet syndrome

that benign paralogous variants in our approach are unlikely to provide evidence for benignity (likelihood odds ratios: para-SAME: 1.80, 95% CI 1.76–1.85 and para-DIFF: 1.82, 95% CI 1.79–1.85). Paralog conserved residues are enriched for pathogenic variants [8], whereas benign variants are more likely to occur in protein regions tolerant to variation, which are often not conserved across paralogs. Thus, the presence of a benign variant at a conserved paralogous residue introduces conflicting evidence—functional constraint implies intolerance to variation, yet a benign variant is observed. This contradiction likely limits the reliability of paralogous benign variants as supportive evidence for benign classification in our approach.

Pathogenic variants in voltage-gated sodium channel (VGSC) genes are associated with a broad spectrum of clinical phenotypes, even within the same gene [4, 30–32]. Prior research demonstrated a strong correlation between different molecular variant effects, such as the gain or loss of a protein function, and the clinical phenotype [33]. We identified phenotypes across VGSC genes with different organ or cellular gene expressions that are caused by corresponding paralogous variants located at the same alignment index position. The location of a variant in the protein structure in VGSC, particularly in critical regions like the selectivity filter or the inactivation gate, is often associated with conserved molecular function [13]. Our findings of 3D-position-based phenotype correlations across VGSC genes likely identify phenotypes caused by variants in paralogous genes with similar molecular effects. The framework we developed assumes that both pathogenicity and the molecular impact of a variant are generally conserved. We confirmed that pathogenicity is often preserved across paralogous genes at conserved residues. Nonetheless, our results suggest that applying correlations derived from the 3D positioning of these variants can potentially identify cases where this conservation does not hold or where variants previously classified as pathogenic were misclassified. The availability of high-quality structural predictions—such as those provided by AlphaFold—can further improve the applicability of 3D variant analyses by providing more complete and reliable protein models. For example, as recently demonstrated, AlphaFold-based predictions enhance the accuracy of interaction interface identification and can yield valuable insights even in regions with low structural resolution [34]. In addition, recent studies provide critical insights into how structural and functional contexts influence variant pathogenicity. Zhou et al. [35] and Cheng et al. [36] demonstrated that disease-associated mutations are enriched at PPI interfaces, emphasizing the importance of structural variant location in understanding variant impacts and its contribution to disease phenotypes.

Despite efforts to standardize criteria for pathogenicity assignment [16] and many improvements in variant interpretation, about 75% of missense variants in ClinVar [17] (accessed 12/2022) are annotated as variants of uncertain significance (VUS). Extending or modifying existing ACMG criteria has been demonstrated as a promising approach to reclassifying VUSs [20, 26, 37–39]. We demonstrated that the PS1 and PM5 criteria of the ACMG guidelines could, in principle, be extended by considering already classified pathogenic variants with corresponding amino acid positions as evidence for pathogenicity. This approach was previously suggested by Clingen Expert curated guidelines for a small set of genes associated with Rasopathies [20]. However, here we have

demonstrated the generalizability of the approach across a large set of 519 gene families and quantified the evidence gained from this approach.

The use of paralogous variants as biologically interpretable evidence in ACMG guidelines requires consideration of key caveats. First, the para-SAME/para-DIFF criteria, which assess individual residue positions, may overlap with approaches that evaluate variant burden across protein regions, such as PM1 ("mutational hotspots"). Several methods can be employed to assess variant burden across protein regions and consequently apply PM1, including using leveraging variants from homologous protein domains [40] or pathogenic-enriched regions [15] (PERs). Para-SAME/para-DIFF criteria may also partially overlap with the PP3 criterium, since it relies on in silico tools (e.g., REVEL [24], BayesDel [41], AlphaMissense [42]), which often incorporate evolutionary conservation as a core feature. Notably, also PM1 and PP3 themselves partially overlap, harboring the risk of evidence double-counting. Recently, to mitigate concerns about double-counting evidence from PP3 with other conservation-based criteria such as PM1, Stenton et al. [43] proposed capping the combined strength of PM1 and PP3 [43]. In line with this recommendation, we suggest implementing a similar cap when combining para-SAME/para-DIFF with PM1 or PP3, to minimize redundancy and ensuring that variant classifications remain appropriately weighted. Second, variants integrated in our framework of pathogenic variants at paralogous positions could be inflated by spliceogenic exonic variants [44]. Although previous results suggest that their impact might be minor on our approach, an exclusion of variants with a predicted high splicing impact could resolve this concern. Third, a limitation of our study is the inclusion of control variants aggregated from the gnomAD database, some of which may be pathogenic despite their presence in the general population. In instances where these control variants are indeed pathogenic, the likelihood ratios calculated in our study may represent underestimations, maintaining the conservative nature of our findings.

In conclusion, our findings suggest that utilizing pathogenic paralogous variants provides significant potential to improve variant interpretation and aid in the diagnosis of pathogenic variants in clinical practice. Reference databases continue to grow and include well-classified pathogenic variants. While we have demonstrated that pathogenic variants in paralogous genes at the same alignment position provide evidence for pathogenicity across all disease-associated gene families, the potential integration of these criteria into the ACMG classification framework would require a careful approach to avoid double counting due to correlation with other criteria that your evolutionary conservation as a feature (e.g., in silico prediction scores). Future iterations of variant interpretation guidelines that consider the presence of paralogous pathogenic variants as evidence of pathogenicity could thus significantly increase the application of criteria based on already established pathogenic variants.

## Conclusions

Our study demonstrates that conserved pathogenic variants in paralogous genes expand the pool of classifiable residues 5.1-fold and raise the odds of pathogenicity 13-fold in a gene-family-specific manner. Cross-paralog correlated phenotypes along the three-dimensional protein structure across the ten voltage-gated sodium channels improves

variant-classification accuracy and highlights common disease mechanisms across *SCN1A, SCN2A, SCN5A,* and *SCN8A*. Together, these findings demonstrate that paralog evidence provides robust, quantifiable support for clinical variant interpretation and should be integrated into future ACMG/AMP guideline refinements. The publicly available pipeline and fully annotated dataset offer an immediate, extensible resource for applying this framework to additional gene families.

## Methods

### Annotation of missense variants from public repositories

#### Missense variants from patients

Patient missense variants were collected from the ClinVar database (ClinVar [17], release December 2024 [45]) and the Human Gene Mutation Database [21] (HGMD®) Professional release 2024.2 [46]. The ClinVar missense variants were obtained in a tabular format from the FTP site (ftp://ftp.ncbi.nlm.nih.gov/pub/clinvar/). To ensure high stringency, only variants classified as "Pathogenic" or "Likely Pathogenic" in their most recent consensus interpretation at the time of data, and that had a ClinVar review status with $\geq 1$ review stars, were included in the primary datasets (see dataset composition below). The HGMD dataset was filtered for "missense variants," "High Confidence" calls (hgmd_confidence = "HIGH" flag), and "Disease causing" state (hgmd_variantType = "DM" flag). All variants were mapped on the human reference genome version GRCh38. Additionally, variants that matched to non-canonical transcripts, as defined by the Matched Annotation from NCBI and EMBL-EBI (MANE) [47, 48], were excluded. For comparisons to the established gene-family–based method for identifying pathogenic enriched regions [15] (PERs) (see the "Comparison to established gene-family-based approaches" section for details), we excluded the pathogenic variants used in defining PERs from the likelihood ratio calculation. In all other analyses, the full set of pathogenic variants was retained.

#### Missense variants from the population gnomAD and Regeneron One Million Exome database

As a comparison group for the likely pathogenic and pathogenic variants from ClinVar and HGMD, we selected missense variants from the general population using data from the Genome Aggregation Database [23] (gnomAD, public release 4.1.0 [49]) and the Regeneron Genetics Center One Million Exome Variant [22] dataset (v1.1.3 [50]). Variants from gnomAD were retrieved in Variant Call Format (VCF) files, and high-quality missense variants were extracted by filtering for entries with the "PASS" flag. Only variants annotated to canonical gene transcripts, as defined by MANE, were included. Missense variants from the Regeneron dataset were also obtained in VCF format. To avoid overlap, all variants present in the gnomAD dataset were excluded from the Regeneron dataset. This resulted in a second independent control dataset comprising mainly rare variants unique to the Regeneron dataset. For comparisons to the established gene-family–based method for identifying pathogenic enriched regions [15] (PERs) (see the "Comparison to established gene-family-based approaches" section for details), we excluded the controls variants used in defining PERs from the likelihood ratio calculation. In all other analyses, the full set of control variants was retained.

### Variant dataset compositions

The current classification of pathogenic and benign variants in public datasets is imperfect, with both inclusive and conservative strategies offering distinct benefits and limitations. A detailed overview of the composition of the datasets used in this study and of the overlap between ClinVar classifications and HGMD classifications is provided in Additional file 1: Fig. S4. To evaluate the impact of different inclusion criteria for pathogenic and control datasets on classification performance, we generated multiple dataset combinations. The most stringent curated dataset (primary dataset) was used for all primary analyses, while the remaining datasets (datasets 2–5) were used for to assess the robustness of our findings under different variant selection criteria. The variant compositions are as follows:

1. Primary dataset: ClinVar "Pathogenic" variants ($\geq 1$ review star) vs. the Regeneron exome dataset (non-overlapping with gnomAD).
2. ClinVar "Likely Pathogenic" and "Pathogenic" variants plus HGMD vs. gnomAD,
3. ClinVar "Likely Pathogenic" and "Pathogenic" variants vs. gnomAD,
4. HGMD variants vs. gnomAD
5. ClinVar "Pathogenic" variants ($\geq 1$ review star) vs. gnomAD

### Annotation of missense variants and associated phenotypes for the voltage-gated sodium channels

### Brain-related phenotypes

We aggregated published patient missense variants in voltage-gated sodium channel genes (VGSC) genes from the literature. All patient variants for *SCN1A* were obtained from Brunklaus et al. (2022) [51] and Brunklaus et al. (2022) [30, 51]. Variants for *SCN2A* were obtained from Wolff et al. [31] and Crawford et al. (2021) [52]. Variants for *SCN3A* were obtained from Zaman et al. (2018) [32]. All *SCN8A* variants are taken from Johannesen et al. (2021) [4]. Affected individuals were recruited through a network of collaborating clinicians, as well as GeneMatcher [53], using a standardized phenotyping sheet to assess clinical characteristics cognition), EEG, neuroimaging, and retrospective data on antiepileptic treatment.

### Non-brain phenotypes

SCN5A variants were obtained from the studies conducted by Milman et al. (2021) [3] and Walsh et al. (2021) [54]. Data from SCN4A, SCN9A, SCN10A, and SCN11A variants were collected from various publications listed in Table S2. Variants in the VGSC genes that were not missense-constrained were filtered for the maximum population frequency (MAF). We inferred the MAF thresholds by using the approach described by Whiffin et al. (2017) [55], via the authors' app (https://www.cardiodb.org/allelefrequency app), based on the phenotype's estimated prevalence, mode of inheritance, and penetrance of the phenotype. We categorized SCN4A variants related to myotonia congenita and paramyotonia congenita and SCN9A variants related to primary erythromelalgia and paroxysmal episodic pain disorder into single categories (relaxation impairment

disorders and paroxysmal pain disorders, respectively) based on their shared molecular pathology and pathophysiology after applying the MAF filter.

We mapped all variants to their Ensembl canonical transcript [47] (*SCN1A*: ENST00000303395, *SCN2A*: ENST00000283256, *SCN3A*: ENST00000283254, *SCN4A*: ENST00000435607, *SCN5A*: ENST00000423572, *SCN8A*: ENST00000283254, *SCN9A*: ENST00000409672, *SCN10A*: ENST00000449082, *SCN11A*: ENST00000302328). Only phenotypes associated with variants at more than five different protein positions were considered. The original and harmonized phenotype annotations for each phenotype are listed in Table S2.

### Gene family definition

We obtained the paralogous genes that belong to a gene family from Pérez-Palma et al. (2020) [15], as originally described in Lal et al. (2020) [8]. Briefly, the human paralog definitions were taken from Ensembl BioMart [56, 57] and filtered for those with an HGNC symbol [7]. For each gene, the canonical transcript as defined by MANE was considered. To avoid aligning highly diverged sequences, families with less than 80% similarity on the full protein sequence were removed.

### Definition of paralogous variants

Paralogous genes arise from gene duplication events of a common ancestral gene and typically share a high degree of sequence and structural similarity, often performing similar functions. We define paralogous variants as those located at corresponding positions in paralogous genes (i.e., the same alignment index in the sequence alignment and the same reference amino acid). For all the protein sequences within the same gene family, we performed a multiple sequence alignment using the biostrings [58] and msa [59] R libraries. We then mapped pathogenic and general population variants onto these multiple sequence alignments. Given two variants on two different genes of the same gene family, we considered them as paralogous variants if they satisfied the two following conditions: (1) they are located at the same position in the multiple protein sequence alignment of the gene family, and (2) the reference amino acid in the target gene and the paralogous gene is the same (Additional file 1: Fig. S5).

We further establish an expanded set of criteria, termed para-SAME and para-DIFF, which is defined as follows:

> para-SAME: This refers to a pathogenic paralogous variant that exhibits the same amino acid substitution as the investigated variant.
> para-DIFF: This denotes a pathogenic paralogous variant that exhibits a different amino acid substitution compared to the investigated variant.

### Calculation of the positive likelihood ratio when a pathogenic paralogous variant is found

For each gene, we calculated the positive likelihood ratio using our aggregated set of pathogenic and general population variants for the para-SAME/DIFF criteria (Additional file 1: Fig. S5). While considering the definition of the criteria (see above) we counted for each gene (i) the number of pathogenic variants for which at least one

pathogenic paralogous variant was observed and (ii) the number of pathogenic variants for which no pathogenic paralogous variant was observed. For the same gene we also counted (i) the number of control variants for which at least one pathogenic paralogous variant was observed and (ii) the number of control variants for which no pathogenic paralogous variant was observed. To determine the level of evidence each criterion can define we calculated the positive likelihood ratios for the two cases: (A) Presence of a pathogenic paralogous variant with either the same amino acid substitution (para-SAME) and (B) Presence of a pathogenic paralogous variant with a different amino acid substitution (para-DIFF). The positive likelihood ratio was computed using the sensitivity and specificity of the test:

$$\text{Positive likelihood ratio}(\text{LR}+) = \frac{\text{Sensitivity}}{\left(1 - \text{Specificity}\right)} = \frac{\left(\frac{\text{TP}}{\text{TP}+\text{FN}}\right)}{1 - \left(\frac{\text{TN}}{\text{TN}+\text{FP}}\right)} \qquad (1)$$

where LR + represents the positive likelihood ratio, TP (true positives) denotes the number of pathogenic variants, for which a pathogenic variant is observed at a conserved corresponding paralogous residue position, TN (true negatives) indicates the number of variants from the general population, for which no pathogenic variants is observed at a corresponding paralogous residue position, FP (false positives) represents the number of population variants, for which a pathogenic variant is observed at a conserved corresponding paralogous residue position, and FN (false negative) denotes the number of pathogenic variants observed, for which no pathogenic variant is observed at a corresponding paralogous residue position. We calculated the LR + both individually for each gene as well as combined across all genes. For the gene-wise metric, we counted the variants denoting TP, FP, TN, and FN for each gene separately. For the combined metric we assessed the numbers for TP, FP, TN, and FN across all disease-associated genes within a gene family together to end up with a single LR +. All analyses were performed using R v.4.2.1.

### Comparison to established gene-family-based approaches

To compare our results to an established gene-family-based approach which identified pathogenic enriched regions (PERs) across paralogous genes on an exome-wide scale [15], we gathered an independent set of variants (see the " Annotation of missense variants from public repositories" section) which was not previously used nor in the PER approach and we repeated the calculation outlined above.

To estimate LR + that are not mediated by paralog conservation we repeated the analysis described above for three paralog conservation sub-groups using the Parazscore [8]. The groups we considered are alignment positions with gene family wise (1) maximum Parazscore, indicating full paralog conservation across the gene family at the alignment position; (2) Parazscore > 0 and not maximum Parazscore, indicating high paralog conservation at this alignment position but not full conservation; and (3) Parazscore < 0, indicating low levels of conservation between paralogous genes at the alignment position.

### Identification of phenotype correlation based on 3D-variant positions

To identify phenotypes associated with variants located at corresponding positions across voltage-gated sodium channels (VGSCs), we evaluated the spatial distribution of sodium channel protein structures for variants associated with each phenotype. We tallied the number of patients reported for each variant in every phenotype. Since not all sodium channels had available protein structures, each patient variant from each sodium channel gene was mapped onto the SCN2A protein sequence through the multiple sequence alignment of the whole gene family. Then the variants were mapped on the Nav1.2 (SCN2A) protein structure using its corresponding Uniprot amino acid position. The Euclidean distance between the Cα (alpha-carbon) atoms of all amino acids in the structure was computed, and residues falling within this threshold were designated as spatially proximate neighbors. For each residue, we tallied the number of patients carrying a variant in either the residue itself or any of its spatially proximal neighbors. These counts were stratified by phenotype, generating a phenotype-specific 3D-variant distribution. To quantify phenotype similarity based on shared 3D-variant positions, we computed Kendall correlation coefficients between all phenotype-specific distributions. The statistical significance of these correlations was assessed using Bonferroni correction ($P < 0.05$) to adjust for multiple comparisons.

### Integrating variant similarity between phenotypes for the assessment of paralogous variant-based pathogenicity

We next explored whether utilizing phenotype correlation could refine the selection of variants for our paralogous patient variant approach. To test our hypothesis we first extracted the variants of the most common phenotypes in each sodium channel with > 40 different variants (*SCN1A*: Dravet Syndrome, *SCN2A*: Early onset developmental epileptic encephalopathy (DEE), *SCN5A*: Brugada Syndrome, *SCN8A*: DEE). We divided these cohorts randomly into four subsets of patient variants, each containing 25% of the variants. We then combined three of the four subsets (representing 75% of variants for each phenotype) with our remaining patient cohort containing all variants associated with other phenotypes. Following the approach outlined in the previous section we then identified 3D-variant position-based phenotype correlations. Finally, using the independent test cohort (the fourth subset), we calculated the LR + of patient vs control variants (a) using paralogous pathogenic variants associated with non-correlated phenotypes and (b) using paralogous pathogenic variants with significant (Bonferroni adjusted $P < 0.05$) 3D-position-based phenotype correlation. We repeated this approach three times, such that each set of variants was used as part of the training set three times and once as the test set, and calculated the LR + by summing up the individual TP, FP, TN, and FN values of each iteration.

### Supplementary Information

---

Additional file 1: Contains all supplementary figures S1 to S5.

Additional file 2. Table S1: List of amino acid residues overlapping with paralogous (likely)-pathogenic variants from ClinVar ≥ 1 review star.

Additional file 3. Table S2: List of patients from literature with 22 distinct phenotypes associated with genetic variants in voltage-gated sodium channels.

Additional file 4: Peer review history.

**Acknowledgements**
Not applicable.

**Review history**
The review history is available as Additional file 4.

**Peer review information**

**Authors' contributions**
Conceptualization: T.B., A.I., D.L. I.H.; data curation: T.B., A.I., E.P; analysis: T.B, A.I.; supervision: D.L., P.M., M.N.; writing—original draft: T.B., A.I., L.M.; writing—editing: D.L., P.M., S.C, L.S., S.P., L.M.

**Funding**
Funding for this work was provided by the German Federal Ministry for Education and Research (BMBF, Treat-ION, 01GM1907D) to D.L., T.B., and P.M., by the BMBF (Treat-Ion2, 01GM2210B) to P.M, the Fonds Nationale de la Recherche in Luxembourg (FNR, Research Unit FOR-2715, INTER/DFG/21/16394868 MechEPI2) to P.M., the Agencia Nacional de Investigación y Desarrollo de Chile (ANID, Fondecyt 1221464 grant) to E.P., the Familie SCN2A foundation 2020 Action Potential Grant to E.P., the Dravet Syndrome Foundation (grant number, 272016) to D.L, and the NIH NINDS (Channelopathy-Associated Epilepsy Research Center, 5-U54-NS108874) to D.L.

**Data availability**
ClinVar missense-variant dataset, December 2024 release (NCBI) [45].
HGMD Professional variant dataset, Release 2024.2 (QIAGEN) [46].
Genome Aggregation Database (gnomAD) variant dataset, v4.1.0 (Broad Institute) [49].
Regeneron Genetics Center One-Million-Exome variant dataset, v1.1.3 (Regeneron Genetics Center) [50].
Matched Annotation from NCBI and EMBL-EBI (MANE) Select & Plus Clinical transcript set, v1.4 (NCBI / EMBL-EBI) [48].
Ensembl BioMart human gene paralogue dataset, release 114 (EMBL-EBI) [57].
All derived summary tables are provided as Supplementary Data. Code to reproduce the analyses is archived on Zenodo under the MIT License (https://doi.org/10.5281/zenodo.15061515) [60] and mirrored on GitHub (https://github.com/TobiasBruenger/paraSAME-DIFF-annotation) [61].

## Declarations

**Ethics approval and consent to participate.**
Not applicable.

**Consent for publication**
Not applicable.

**Competing interests**
The authors report no competing interests.

**Author details**
[1]Cologne Center for Genomics (CCG), University of Cologne, Cologne 50931, Germany. [2]Department of Neurology, Mayo Clinic in Florida, Jacksonville, Fl 32224, USA. [3]Centro de Genética y Genómica, Facultad de Medicina Clínica Alemana, Universidad del Desarrollo, Santiago, Chile. [4]Department of Neurology, The University of Texas Health Science Center at Houston, Houston, TX, USA. [5]Division of Neurology, Children's Hospital of Philadelphia, Philadelphia, PA 19104, USA. [6]The Epilepsy NeuroGenetics Initiative (ENGIN), Children's Hospital of Philadelphia, Philadelphia, PA, USA. [7]Department of Biomedical and Health Informatics (DBHi), Children's Hospital of Philadelphia, Philadelphia, PA 19104, USA. [8]Epilepsy Genetics Program, Division of Epilepsy and Clinical Neurophysiology, Department of Neurology, Boston Children's Hospital, Boston, MA, USA. [9]Department of Neurology, Perelman School of Medicine, University of Pennsylvania, Philadelphia, PA 19104, USA. [10]Luxembourg Centre for Systems Biomedicine, University of Luxembourg, Esch-Sur-Alzette, Luxembourg. [11]Stanley Center of Psychiatric Research, Broad Institute of Harvard and MIT, 75 Ames Street, Cambridge, MA 02142, USA. [12]Epilepsy Center, Neurological Institute, Cleveland Clinic, Cleveland, OH 44106, USA. [13]Center for Neurogenetics, The University of Texas Health Science Center at Houston, Houston, TX, USA.

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

## 