## [Additional file 4: Peer review history. · Genome Biology]

Review history

First round of review

Reviewer 1

Brunger, Ivaniuk and colleagues describe the results of a large-scale data repurposing effort to establish the degree to which the pathogenicity of a missense variant at a conserved amino acid residue in a gene of interest is associated with the pathogenicity of the 'same variant' in a paralogous gene. As acknowledged by the authors, this concept has been appreciated on a small scale for selected genes/gene families in the past, but theirs appears to be the first effort to systematically assess the usefulness of this approach in understanding the pathogenicity of missense variants genome-wide. There are several strengths of this study, including the rigor of the computational analyses and demonstration of proof-of-concept in a specific gene family. Their findings represent a modest but valuable advance in addressing one of the biggest challenges in human genetics (the interpretation of rare missense variants). The following comments and recommendations are intended to strengthen the manuscript and ensure the results can be translated into clinical practice.

Major comments:

1. The merging of all ClinVar LP/P and HGMD "Disease-causing" variants is problematic when defining the 'case variant' set. This reviewer's experience is that HGMD variant classifications are not as robust or as reliable (when assessed using 2024 tools/evidence) as ClinVar. The strength of evidence that informed the ClinVar classification is also not considered. The numbers of variants in each set of the Venn diagram of the union are not shown. The degree to which HGMD variants had conflicting interpretations in ClinVar is not shared (i.e., how many HGMD DMs were LB/B/VUS in ClinVar?).
2. Related to comment #1, the overlap of variants between the ClinVar/HGMD set and the gnomAD set was not clear. Consider the use of additional 'control' variants that would not have been used to inform ClinVar or HGMD interpretations (e.g., RGC Million Exome Variant Browser). The approach of using population database variants as controls is a more reasonable approach for dominant disease genes than for recessive disease genes, and this point was not clear in the manuscript.
3. The use of the terms "para-PS1" and "para-PM5" suggests that the LR+ for these criteria were at the expected level for strong and moderate evidence, respectively. Please clarify if this was the case. If not, consider using a different term or making this explicit.
4. "Double counting" evidence and the presumed non-independence of different criteria is a concern. To what degree does the evidence considered by these authors overlap with "mutational hotspot" evidence (PM1) that might have already been used in variant classification? Please comment on the degree to which this paper (not written by this reviewer) is relevant: Wiel et al., AJHG, 2023, PMID 36563679. To what degree might advanced machine-learning predictive models have already included this information related to paralogous genes, having learned from ClinVar (VARITY) or predicted protein structures (AlphaMissense)?

Minor comments:

5. Some of the data presented by the authors (e.g., Figure 2B) suggests a possible role for evidence of 'benign' variation at a conserved residue in a paralog being associated with the benign-ness of a variant in a gene of interest. Please discuss.

6. In Supplemental Table 1, some residues had more than one supporting paralogous variant. Did the number of such variants correlate with pathogenicity classifications of the variant of interest?

Very minor comments / typos:

- Line 104: Please acknowledge AMP in addition to ACMG when describing the variant interpretation guidelines for the first time
- Line 245: reference '14' should be superscript
- Figure 1: typo in "Brugada syndrome"
- Tab in .xls file for Supplemental Table 2 says Supp Table 3
- Please include unique gene identifier besides gene symbol in Supplemental Table 1

Reviewer 2

This study presented an integrative structural biology framework to explore functional consequences of disease-associated mutations from whole-exome sequencing studies. Specifically, they mapped 2.5 million pathogenic and general population variants from the ClinVar, HGMD, and gnomAD databases onto a total of 9,990 genes. The authors showed that the presence of a pathogenic variant in a paralogous gene is associated with elevated ratios (8.32) for variant pathogenicity. Using ten genes encoding voltage-gated sodium channels and 22 expert-curated disorders, the authors identified cross-paralog correlated phenotypes based on 3D structural spatial position. Overall, this is a comprehensive study to explore paralogous variants from protein structural perspectives. The manuscript is well-structured and written as well. Several specific minor comments should be discussed and considered further as below.

The authors are suggested to give more explanations about paralogous variants.

It is unclear paralogous variants are common variants or more likely to be rare variants. The authors should provide allele frequency for paralogous variants.

The authors should provide more details about how to conduct 3D spatial analysis for 3D-variant positions. It is unclear whether large-scale predicted 3D protein structural information, such as AlphaFold2/3 may increase the accuracy of the current 3D-variant position analyses. More explanations and discussion should be provided.

Several related references should be discussed as well: PMID: 33514395; PMID: 33558758; PMID: 39448882

The authors are suggested to provide the communities about how to freely access of the data.

Authors' response to reviewers

Editor:

0.1: We ask that the version of source code used in the manuscript is deposited in a DOI-assigning repository, such as Zenodo, with the link also included, in addition to GitHub. Please note that if we decide to publish your manuscript, we will require that the source code is made publicly available under an open source license compliant with Open Source Initiative. All this information should be listed in a separate Availability of Data and Materials section of the manuscript.

Response 0.1:

We have now deposited the scripts and dataset in Zenodo, with the DOI: <https://doi.org/10.5281/zenodo.15061514>. A reference to this resource and its open-source license has been added to the "Availability of Data and Materials" section.

Updated Availability of Data and Materials section:

Data is available in the Supplementary Tables. Code to annotate para-SAME/DIFF criteria with a custom variant dataset for any gene family and the scripts to reproduce the analysis and visuals of the manuscript can be obtained from <https://doi.org/10.5281/zenodo.15061514>.

Reviewer reports:

Reviewer #1: Brunger, Ivaniuk and colleagues describe the results of a large-scale data repurposing effort to establish the degree to which the pathogenicity of a missense variant at a conserved amino acid residue in a gene of interest is associated with the pathogenicity of the 'same variant' in a paralogous gene. As acknowledged by the authors, this concept has been appreciated on a small scale for selected genes/gene families in the past, but theirs appears to be the first effort to systematically assess the usefulness of this approach in understanding the pathogenicity of missense variants genome-wide. There are several strengths of this study, including the rigor of the computational analyses and demonstration of proof-of-concept in a specific gene family. Their findings represent a modest but valuable advance in addressing one of the biggest challenges in human genetics (the interpretation of rare missense variants). The following comments and recommendations are intended to strengthen the manuscript and ensure the results can be translated into clinical practice.

Major comments:

Reviewer 1

1.1: The merging of all ClinVar LP/P and HGMD "Disease-causing" variants is problematic when defining the 'case variant' set. This reviewer's experience is that HGMD variant classifications are not as robust or as reliable (when assessed using 2024 tools/evidence) as ClinVar. The strength of evidence that informed the ClinVar classification is also not considered. The numbers of variants in each set of the Venn diagram of the union are not shown. The degree to which HGMD variants had conflicting interpretations in ClinVar is not shared (i.e., how many HGMD DMs were LB/B/VUS in ClinVar?).

Response 1.1:

We thank the reviewer for highlighting the concern about merging ClinVar and HGMD variants. We acknowledge the concerns about variant classification quality from HGMD. To enhance reliability and enrich our dataset for truly disease-associated variants, our primary analysis now exclusively includes ClinVar Likely Pathogenic (LP) and Pathogenic (P) with ≥ 1 review stars.

Additionally, to maximize the confidence in control variants being truly benign, we selected variants from the Regeneron One Million Exome dataset that are not present in gnomAD 4.1.0 as the control set (see also response to comment 1.2). This ensures higher reliability of the variant classifications as the gnomAD data is likely used in ClinVar variant classification (directly or indirectly through scores that have been trained/calibrated using this data).

To ensure our results reflect the latest data, we have updated all variant datasets to their most recent versions, including ClinVar (December 2024), gnomAD (4.1.0), HGMD Professional (2024.2), and Regeneron One Million Exomes (v1.1.3).

We have updated the Methods section to reflect the integration of the new dataset and the expanded variant selection process:

Missense variants from patients

Patient missense variants were collected from the ClinVar database (ClinVar, release December 2024) and the Human Gene Mutation Database (HGMD® Professional release 2024.2). The ClinVar missense variants were obtained in a tabular format from the FTP site (<ftp://ftp.ncbi.nlm.nih.gov/pub/clinvar/>). To ensure high stringency, only variants classified as "Pathogenic" or "Likely Pathogenic" in their most recent consensus interpretation at the time of data, and that had a ClinVar review status with ≥ 1 review stars, were included in the primary datasets (see dataset composition below). The HGMD dataset was filtered for "missense variants," "High Confidence" calls (hgmd_confidence = "HIGH" flag), and "Disease causing" state (hgmd_variantType = "DM" flag). All variants were mapped on the human reference genome version GRCh38. Additionally, variants that matched to non-canonical transcripts, as defined by the Matched Annotation from NCBI and EMBL-EBI (MANE), were excluded. For comparisons to the established gene-family-based method for identifying pathogenic enriched regions (PERs) (see "Comparison to Established Gene-Family-Based Approaches" for details), we excluded the pathogenic variants used in defining PERs from the likelihood ratio calculation. In all other analyses, the full set of pathogenic variants was retained.

Missense variants from the population gnomAD and Regeneron One Million Exome database

As a comparison group for the likely pathogenic and pathogenic variants from ClinVar and HGMD, we selected missense variants from the general population using data from the Genome Aggregation Database (gnomAD, public release 4.1.0) and the Regeneron Genetics Center One Million Exome Variant dataset (v1.1.3). Variants from gnomAD were retrieved in Variant Call Format (VCF) files, and high-quality missense variants were extracted by filtering for entries with the "CSQ" field and "PASS" flag. Only variants annotated to canonical gene transcripts, as defined by MANE, were included. Missense variants from the Regeneron dataset were also obtained in VCF format. To avoid overlap, all variants present in the gnomAD dataset were excluded from the Regeneron dataset. This resulted in a second independent control dataset comprising mainly rare variants unique to the Regeneron dataset. For comparisons to the established gene-family-based method for identifying pathogenic enriched regions¹⁵ (PERs) (see "Comparison to Established Gene-Family-Based Approaches" for details), we excluded the controls variants used in defining PERs from the likelihood ratio calculation. In all other analyses, the full set of control variants was retained.

Variant dataset compositions

The current classification of pathogenic and benign variants in public datasets is imperfect, with both inclusive and conservative strategies offering distinct benefits and limitations. A detailed overview of the composition of the datasets used in this study and of the overlap between ClinVar classifications and HGMD classifications is provided in Supplementary Figure 2. To evaluate the impact of different inclusion criteria for pathogenic and control datasets on classification performance, we generated multiple dataset combinations. The most stringent curated dataset (Dataset 1) was used for all primary analyses, while the remaining datasets (Datasets 2–5) were used to assess the robustness of our findings under different variant selection criteria. The variant compositions are as follows:

1. **Main dataset:** ClinVar "Pathogenic" variants (≥ 1 review star) vs. the Regeneron exome dataset (non-overlapping with gnomAD).
2. ClinVar "Likely Pathogenic" and "Pathogenic" variants plus HGMD vs. gnomAD,
3. ClinVar "Likely Pathogenic" and "Pathogenic" variants vs. gnomAD,
4. HGMD variants vs. gnomAD
5. ClinVar "Pathogenic" variants (≥ 1 review star) vs. gnomAD

To ensure transparency and address concerns about the overlap between HGMD and ClinVar variant classifications, we added Supplementary Figure 2, which provides a detailed breakdown of the number of variants in each dataset and explicitly illustrates the distribution of HGMD "Disease-causing" (DM) variants overlapping with ClinVar LB/B/VUS classifications.

Expansion of the Methods section and addition of Supplementary Figure 3:

The current classification of pathogenic and benign variants in public datasets is imperfect, with both inclusive and conservative strategies offering distinct benefits and limitations. A detailed overview of the composition of the datasets used in this study and of the overlap between ClinVar classifications and HGMD classifications is provided in Supplementary Figure 2. To evaluate the impact of different inclusion criteria for pathogenic and control datasets on classification performance, we generated multiple dataset combinations. The most stringent curated dataset (Dataset 1) was used for all primary analyses, while the remaining datasets (Datasets 2–5) were used to assess the robustness of our findings under different variant selection criteria. The variant compositions are as follows:

Supplementary Figure 2: Overview of variant datasets and classification overlap. A) The number of variants included in the study from patient and population control datasets. Pathogenic variants were derived from ClinVar (all "Likely Pathogenic/Pathogenic" [Lp/P], see methods for details) and the Human Gene Mutation Database (HGMD, "Disease-causing" [DM] variants). Population control variants were sourced from gnomAD (v4.1.0) and the Regeneron Genetics Center One Million Exome Variant dataset (v.1.1.3, non-overlapping with gnomAD). B) Overlap of HGMD variants with ClinVar classifications. Variants are categorized as "Likely Pathogenic/Pathogenic" (Lp/P), "Not present in ClinVar," "Variant of Uncertain Significance" (VUS), and "Likely Benign/Benign" (Lb/B). The total number of variants for each category is shown above the bars.

We updated the Results section to clarify that incorporating the latest dataset versions and stringent variant filtering did not lead to significant changes in the observed likelihood ratios (LR+). Specifically, the para-SAME LR+ changed from 8.0 to 13.0, and the para-DIFF LR+ increased slightly from 4.3 to 6.0, confirming the stability of our findings.

These changes are reflected in the Results section and Figure 2B:

Next, we quantified the value of incorporating pathogenic variants at paralogous positions to assess the pathogenicity of novel variants. Therefore, in addition to the aforementioned pathogenic variants, we included 550,251 variants from the Regeneron One Million Exome database that were not present in gnomAD, which served as controls in our study. When a pathogenic paralogous variant with the

same amino acid exchange was present at a corresponding alignment index position (termed the **para-SAME criterion**, for details on the approach, see Methods), we observed across 414 gene families an average **LR+ of 13 (12.5-13.7, 95% confidence interval [CI]; Figure 2B)**. Notably, even for paralogous variants with a different substitution at the same alignment index position (termed the **para-DIFF criterion**), we observed an LR+ of 6.0 [5.7–6.2, 95% CI].

Figure 2: Individual pathogenic paralogous variants can serve as a proxy for variant pathogenicity. A) Number of amino acid residues in 414 gene families that have a pathogenic variant (ClinVar, one star+) at the same protein position in the same gene or a corresponding protein residue in a paralogous gene. B) Amino acids with a paralogous pathogenic variant at a paralogous alignment position have an increased positive likelihood ratio (LR+ >1). In contrast, amino acids with a paralogous control variant (Regeneron, not present in gnomAD) at a paralogous alignment position are not enriched for pathogenic variants. Each data point represents the gene-wise LR+. The gene-wise LR+ was calculated for genes where 10 or more pathogenic variants (ClinVar, one star+) and control variants (Regeneron, not present in gnomAD) could be mapped.

Additionally, we now demonstrate that LR+ ratios remain stable across different dataset compositions (Supplementary Figure 2), further reinforcing the robustness of our approach.

Expanded result section and new Supplementary Figure 2:

Notably, even for paralogous variants with a different substitution at the same alignment index position (termed the para-PM5 criterion), we observed an **LR+ of 6.0 [5.7–6.2, 95% CI]**. We observe that LR+ values remain highly correlated across different variant set compositions, demonstrating the robustness of our approach, while considering variability in classification quality of pathogenic and control variants (Supplementary Figure 4). However, we observed substantial variation in the observed LR+ across different genes (Figure 2B), indicating that gene-family-specific calibration is advisable.

Supplementary Figure 4. Pearson correlation matrix of LR+ values across five different dataset compositions. Each cell shows the correlation between the LR+ estimates for pathogenic variants for the para-DIFF and para-SAME criterium: 1) ClinVar pathogenic variants (all review stars) plus HGMD vs gnomAD, 2) ClinVar pathogenic variants (all review stars) vs gnomAD, 3) HGMD variants vs gnomAD, 4) ClinVar pathogenic variants (≥ 1 review star) vs gnomAD, 5) ClinVar pathogenic variants (≥ 1 review star) vs a Regeneron exome dataset non-overlapping with gnomAD. High correlations in the heatmap indicate that the LR+ estimates remain consistent across all dataset comparisons. **: significant at $p < 0.001$ after Bonferroni multiple testing correction.

1.2: Related to comment #1, the overlap of variants between the ClinVar/HGMD set and the gnomAD set was not clear. Consider the use of additional 'control' variants that would not have been used to inform ClinVar or HGMD interpretations (e.g., RGC Million Exome Variant Browser). The approach of using population database variants as controls is a more reasonable approach for dominant disease genes than for recessive disease genes, and this point was not clear in the manuscript.

Response 1.2:

As outlined in **Response 1.1**, we have addressed this concern by expanding variant sets, including population variants. While our results remain largely unchanged (see updated Results section in **Response 1.1**), we now provide greater granularity on effect sizes. We thank the reviewer for this suggestion.

1.3: The use of the terms "para-PS1" and "para-PM5" suggests that the LR+ for these criteria were at the expected level for strong and moderate evidence, respectively. Please clarify if this was the case. If not, consider using a different term or making this explicit.

Response 1.3:

We agree with the reviewer's suggestion. To avoid confusion about evidence strength, we have replaced the terms *para-PS1* and *para-PM5* with *para-SAME* and *para-DIFF*, respectively, throughout the manuscript and updated the Method section, where these

terms are first introduced.

Updated method section:

Paralogous genes arise from gene duplication events of a common ancestral gene and often share a high degree of sequence similarity and similar or identical functions. We refer to paralogous variants as variants located at corresponding positions in these genes (i.e., the same alignment index and the same reference amino acid in a multiple sequence alignment). For all the protein sequences within the same gene family, we performed a multiple sequence alignment using the `biostrings` and `msa` R libraries. We then mapped pathogenic and general population variants onto these multiple sequence alignments. Given two variants on two different genes of the same gene family, we considered them as paralogous variants if they satisfied the two following conditions: (1) they are located at the same position in the multiple protein sequence alignment of the gene family, and (2) the reference amino acid in the target gene and the paralogous gene is the same (Supplementary Figure 1).

We further establish an expanded set of criteria, termed **para-DIFF** and **para-SAME**, which is defined as follows:

- **para-SAME:** This denotes a pathogenic paralogous variant that exhibits the same amino acid substitution as the investigated variant.
- **para-DIFF:** This denotes a pathogenic paralogous variant that exhibits a different amino acid substitution compared to the investigated variant.

1.4: "Double counting" evidence and the presumed non-independence of different criteria is a concern. To what degree does the evidence considered by these authors overlap with "mutational hotspot" evidence (PM1) that might have already been used in variant classification? Please comment on the degree to which this paper (not written by this reviewer) is relevant: Wiel et al., *AJHG*, 2023, PMID 36563679. To what degree might advanced machine-learning predictive models have already included this information related to paralogous genes, having learned from ClinVar (VARITY) or predicted protein structures (AlphaMissense)?

Response 1.4:

We appreciate the reviewer's comments regarding the potential risk of double counting evidence, a well-recognized challenge when integrating multiple ACMG criteria. The information utilized for our approach (para-SAME/para-DIFF) may partially overlap with data and methodologies used to define mutational hotspots (PM1), such as homologous domains such as those identified by Wiel et al. (2023) or are features in computational *in-silico* prediction tools (PP3).

To guide the reader that our residue level approach is not independent from domain-based annotation or computational scores, we have expanded the Discussion to explicitly highlight these conceptual overlaps and outline mitigation strategies to reduce

the risk of double counting evidence.

Expanded discussion section:

The use of paralogous variants as biologically interpretable evidence in ACMG guidelines requires consideration of key caveats. Notably, the para-SAME/para-DIFF criteria, which assess individual residue positions, may overlap with approaches that evaluate variant burden across protein regions, such as PM1 (“mutational hotspots”). Several methods can be employed to assess variant burden across protein regions and consequently apply PM1, including using leveraging variants from homologous protein domains (Wiel et al. 2023) or pathogenic-enriched regions (PERs, Pérez-Palma et al. 2020). Para-SAME/para-DIFF criteria may also partially overlap with the PP3 criterium, since it relies on *in-silico* tools (e.g., REVEL, BayesDel, AlphaMissense), which often incorporate evolutionary conservation as a core feature. Notably, also PM1 and PP3 themselves partially overlap, harboring the risk of evidence double-counting. Recently, to mitigate concerns about double-counting evidence from PP3 with other conservation-based criteria such as PM1, Stenton et al. (2024) proposed capping the combined strength of PM1 and PP3. In line with this recommendation, we suggest implementing a similar cap when combining para-SAME/para-DIFF with PM1 or PP3, to minimize redundancy and ensuring that variant classifications remain appropriately weighted.

Minor comments:

1.5: Some of the data presented by the authors (e.g., Figure 2B) suggests a possible role for evidence of 'benign' variation at a conserved residue in a paralog being associated with the benign-ness of a variant in a gene of interest. Please discuss.

Response 1.5:

We thank the reviewer for this insightful comment. While benign variation at paralogous positions may, in some cases, suggest reduced constraint and potential benignity, our analyses indicate that this is not a broadly applicable principle. Specifically, the likelihood ratios for para-SAME (1.80, 95% CI 1.76–1.85) and para-DIFF (1.82, 95% CI 1.79–1.85) suggest that paralogous benign variants do not provide strong predictive value for classifying novel variants as benign.

To address this, we have expanded the Discussion section to outline why benign variation at conserved residues in paralogs does not serve as evidence to support variant benignity:

Pathogenic missense variants in paralogous genes can serve as a proxy for pathogenicity. Within a protein sequence, pathogenic variants are unevenly distributed and tend to accumulate in certain regions that are critical for protein function. These pathogenic variant-enriched regions have proven valuable for variant classification through established guidelines for variant interpretation and the use of *in-silico* prediction algorithms. Moreover, the observation that critical protein regions tend to be evolutionarily conserved between paralogous

genes can be harnessed to enhance statistical robustness by incorporating pathogenic variants across these paralogous genes. Still, about 70% of pathogenic variants are located outside the regions identified as essential. As a result, individual pathogenic variants in paralogous genes outside these regions were not considered for variant interpretation. In a study examining long QT syndrome, it was observed that individual pathogenic variants in paralogous genes are often located at paralogous positions as determined from multiple sequence alignments, suggesting that the presence of a pathogenic variant at a particular position may serve as a proxy of pathogenicity at that alignment position in other paralogs. While pathogenic paralogous variants provide strong evidence for pathogenicity, we showed that benign paralogous variants in our approach are unlikely to provide evidence for benignity (Likelihood odds ratios: para-SAME: 1.80, 95% CI 1.76–1.85 and para-DIFF: 1.82, 95% CI 1.79–1.85). Paralog conserved residues are enriched for pathogenic variants (Lal et al., 2020), whereas benign variants are more likely to occur in protein regions tolerant to variation, which are often not conserved across paralogs. Thus, the presence of a benign variant at a conserved paralogous residue introduces conflicting evidence—functional constraint implies intolerance to variation, yet a benign variant is observed. This contradiction likely limits the reliability of paralogous benign variants as supportive evidence for benign classification in our approach.

1.6: In Supplemental Table 1, some residues had more than one supporting paralogous variant. Did the number of such variants correlate with pathogenicity classifications of the variant of interest?

Response 1.6:

We thank the reviewer for this insightful question. Our data confirm that the number of pathogenic paralogous variants at the same residue is associated with the pathogenicity classification of the variant of interest (Supplementary Figure 4).

To highlight this key finding, we expanded the Results section and introduced Supplementary Figure 4.

We observe that LR+ values remain highly correlated across different variant set compositions, demonstrating the robustness of our approach, while considering variability in classification quality of pathogenic and control variants (Supplementary Figure 4). However, we observed substantial variation in the observed LR+ across different genes (Figure 2B), indicating that gene-family-specific calibration is advisable. We further investigated whether the presence of multiple pathogenic paralogous variants at the same conserved residue increases the evidence for pathogenicity. As shown in Supplementary Figure 5, a single pathogenic variant in a paralogous gene is associated with a 2.84-fold enrichment of pathogenic variants compared to benign variants at that residue. The fold enrichment of pathogenic variant progressively increases with a higher number of pathogenic paralogous variants.

Supplementary Figure 5: The likelihood that a variant is pathogenic increases with the number of paralogous pathogenic variants at the same position. The bar plots show the ratio of the number of likely pathogenic (LP) and pathogenic (P) variants relative to i) likely benign (LB) and benign (B) variants (left panel) and ii) variants of uncertain significance (VUS) (right panel). This ratio is computed separately for positions where 1, 2, 3, or 4+ paralogous pathogenic variants have been observed (x-axis). The total number of LP/P and LB/B variants (left panel) and of LP/P and VUS variants (right panel) in each category is displayed above the corresponding bars. A higher ratio indicates a greater enrichment of LP/P variants relative to LB/B or VUS variants at the same protein residue. Data were obtained from ClinVar (accessed December 2024).

Very minor comments / typos:

1.7: Line 104: Please acknowledge AMP in addition to ACMG when describing the variant interpretation guidelines for the first time

Response 1.7:

We have updated the text in the introduction to acknowledge AMP in addition to ACMG:

To standardize variant interpretation, the American College of Medical Genetics and Genomics (ACMG) and **the Association for Molecular Pathology (AMP)** and published recommendations for evaluating the variant pathogenicity.

1.8: Line 245: reference '14' should be superscript

Response 1.8:

We have changed reference 14 to superscript.

1.9: Figure 1: typo in "Brugada syndrome"

Response 1.9:

We have corrected the typo to "Brugada syndrome."

1.10: Tab in .xls file for Supplemental Table 2 says Supp Table 3

Response 1.10:

We have corrected the table label in Supplemental Table 2.

1.11: Please include unique gene identifier besides gene symbol in Supplemental Table 1

Response 1.11:

We have added the official HGNC identifiers for each gene in Supplemental Table 1.

Reviewer #2: This study presented an integrative structural biology framework to explore functional consequences of disease-associated mutations from whole-exome sequencing studies. Specifically, they mapped 2.5 million pathogenic and general population variants from the ClinVar, HGMD, and gnomAD databases onto a total of 9,990 genes. The authors showed that the presence of a pathogenic variant in a paralogous gene is associated with elevated ratios (8.32) for variant pathogenicity. Using ten genes encoding voltage-gated sodium channels and 22 expert-curated disorders, the authors identified cross-paralog correlated phenotypes based on 3D structural spatial position. Overall, this is a comprehensive study to explore paralogous variants from protein structural perspectives. The manuscript is well-structured and written as well. Several specific minor comments should be discussed and considered further as below.

Reviewer 2

2.1: The authors are suggested to give more explanations about paralogous variants.

Response 2.1:

We thank the reviewer for highlighting the need to clarify paralogous variants. We have expanded the "Definition of Paralogous Variants" section in the Methods to provide a clearer explanation of how variants in one gene can inform the impact of corresponding variants in a paralog.

Expanded Method section:

Definition of paralogous variants

Paralogous genes arise from gene duplication events of a common ancestral gene and typically share a high degree of sequence and structural similarity, often performing similar functions. We define paralogous variants as those located at corresponding positions in paralogous genes (i.e., the same alignment index in the sequence alignment and the same reference amino acid).

For all the protein sequences within the same gene family, we performed a multiple sequence alignment using the biostrings and msa R libraries. We then mapped pathogenic and general population variants onto these multiple sequence alignments. Given two variants on two different genes of the same gene family, we considered them as paralogous variants if they satisfied the two following conditions: (1) they are located at the same position in the multiple protein sequence alignment of the

gene family, and (2) the reference amino acid in the target gene and the paralogous gene is the same (Supplementary Figure 1).

We further establish an expanded set of criteria, termed **para-SAME** and **para-DIFF**, which is defined as follows:

- **para-SAME:** This refers to a pathogenic paralogous variant that exhibits the same amino acid substitution as the investigated variant.
- **para-DIFF:** This denotes a pathogenic paralogous variant that exhibits a different amino acid substitution compared to the investigated variant.

2.2: It is unclear paralogous variants are common variants or more likely to be rare variants. The authors should provide allele frequency for paralogous variants.

Response 2.2:

We thank the reviewer for this feedback and agree that allele frequency data provides valuable insights into the significance of paralogous variants. We have now included allele frequencies from gnomAD for all pathogenic paralogous variants in Supplementary Table 1.

2.3: The authors should provide more details about how to conduct 3D spatial analysis for 3D-variant positions. It is unclear whether large-scale predicted 3D protein structural information, such as AlphaFold2/3 may increase the accuracy of the current 3D-variant position analyses. More explanations and discussion should be provided.

Response 2.3:

We have provided the script on how to conduct 3D spatial analyses in our Zenodo repository (<https://doi.org/10.5281/zenodo.15061514>).

In addition, we have revised and expanded the Methods section to provide a more detailed description of the 3D spatial analysis, including the mapping of variants to protein structures and the assessment of phenotype correlations:

To identify phenotypes associated with variants located at corresponding positions across voltage-gated sodium channels (VGSCs), we evaluated the spatial distribution of sodium channel protein structures for variants associated with each phenotype. We tallied the number of patients reported for each variant in every phenotype. Since not all sodium channels had available protein structures, each patient variant from each sodium channel gene was mapped onto the SCN2A protein sequence through the multiple sequence alignment of the whole gene family. Then the variants were mapped on the Nav1.2 (SCN2A) protein structure using its corresponding Uniprot amino acid position. To account for spatial clustering of variants, we identified residues within a 5Å cutoff radius, a distance threshold previously established in structural variant analyses (Iqbal et al., 2022). The Euclidean distance between the C α (alpha-carbon) atoms of all amino acids in the structure was computed, and residues falling within this threshold were designated as spatially proximate neighbors. For each residue, we tallied the number of patients carrying a variant in either the residue itself or any of its spatially proximal neighbors. These counts were stratified by phenotype,

generating a phenotype-specific 3D-variant distribution. To quantify phenotype similarity based on shared 3D-variant positions, we computed Kendall correlation coefficients between all phenotype-specific distributions. The statistical significance of these correlations was assessed using Bonferroni correction ($P < 0.05$) to adjust for multiple comparisons.

We have also expanded the Discussion to emphasize that high-quality structural predictions (e.g., from AlphaFold) can improve the accuracy of these 3D variant analyses:

Our findings of 3D-position-based phenotype correlations across VGSC genes likely identify phenotypes caused by variants in paralogous genes with similar molecular effects. The framework we developed assumes that both pathogenicity and the molecular impact of a variant are generally conserved. We confirmed that pathogenicity is often preserved across paralogous genes at conserved residues. Nonetheless, our results suggest that applying correlations derived from the 3D positioning of these variants can potentially identify cases where this conservation does not hold or where variants previously classified as pathogenic were misclassified. The availability of high-quality structural predictions—such as those provided by AlphaFold—can further improve the applicability of 3D variant analyses by providing more complete and reliable protein models. For example, as recently demonstrated, AlphaFold-based predictions enhance the accuracy of interaction interface identification and can yield valuable insights even in regions with low structural resolution (Xiong et al). In addition, recent studies provide critical insights into how structural and functional contexts influence variant pathogenicity. Zhou et al. and Cheng et al. demonstrated that disease-associated mutations are enriched at PPI interfaces, emphasizing the importance of structural variant location in understanding variant impacts and its contribution to disease phenotypes.

2.4: Several related references should be discussed as well: PMID: 33514395; PMID: 33558758; PMID: 39448882

Response 2.4:

We thank the reviewer for these valuable reference suggestions. We have incorporated and discussed findings from PMID: 33514395, PMID: 33558758, and PMID: 39448882 in the Discussion section. These studies provide key insights into structural and functional impacts of variants, protein-protein interactions, and the role of predictive models like AlphaFold in refining 3D variant analysis.

Expanded discussion section (see also Response 2.3):

Our findings of 3D-position-based phenotype correlations across VGSC genes likely identify phenotypes caused by variants in paralogous genes with similar molecular effects. The framework we developed assumes that both pathogenicity and the molecular impact of a variant are generally conserved. We confirmed that

pathogenicity is often preserved across paralogous genes at conserved residues. Nonetheless, our results suggest that applying correlations derived from the 3D positioning of these variants can potentially identify cases where this conservation does not hold or where variants previously classified as pathogenic were misclassified. The availability of high-quality structural predictions—such as those provided by AlphaFold—can further improve the applicability of 3D variant analyses by providing more complete and reliable protein models. For example, as recently demonstrated, AlphaFold-based predictions enhance the accuracy of interaction interface identification and can yield valuable insights even in regions with low structural resolution (Xiong et al). In addition, recent studies provide critical insights into how structural and functional contexts influence variant pathogenicity. Zhou et al. and Cheng et al. demonstrated that disease-associated mutations are enriched at PPI interfaces, emphasizing the importance of structural variant location in understanding variant impacts and its contribution to disease phenotypes.

2.5: The authors are suggested to provide the communities about how to freely access of the data.

Response 2.5:

We have deposited all scripts and datasets in Zenodo under an open-source license (<https://doi.org/10.5281/zenodo.15061514>). The data to use apply para-SAME/DIFF criteria is provided in the Supplementary tables and the Zenodo repository. The "Availability of Data and Materials" section has been updated accordingly (see **Response 0.1**).

Second round of review

Reviewer 1

This will be a useful addition to our field and I hope to see it published

Reviewer 2

The authors have addressed my previous concerns.